# Adapter Naturally Serves as Decoupler for Cross-Domain Few-Shot Semantic Segmentation

**Jintao Tong** [1]   **Ran Ma** [1]   **Yixiong Zou**[✉][1]   **Guangyao Chen** [2]   **Yuhua Li** [1]   **Ruixuan Li** [1]

## Abstract

Cross-domain few-shot segmentation (CD-FSS) is proposed to pre-train the model on a source-domain dataset with sufficient samples, and then transfer the model to target-domain datasets where only a few samples are available for efficient fine-tuning. There are majorly two challenges in this task: (1) the domain gap and (2) fine-tuning with scarce data. To solve these challenges, we revisit the adapter-based methods, and discover an intriguing insight not explored in previous works: the adapter not only helps the fine-tuning of downstream tasks but also naturally serves as a domain information decoupler. Then, we delve into this finding for an interpretation, and find the model's inherent structure could lead to a natural decoupling of domain information. Building upon this insight, we propose the Domain Feature Navigator (DFN), which is a structure-based decoupler instead of loss-based ones like current works, to capture domain-specific information, thereby directing the model's attention towards domain-agnostic knowledge. Moreover, to prevent the potential excessive overfitting of DFN during the source-domain training, we further design the SAM-SVN method to constrain DFN from learning sample-specific knowledge. On target domains, we freeze the model and fine-tune the DFN to learn target-specific knowledge specific. Extensive experiments demonstrate that our method surpasses the state-of-the-art method in CD-FSS significantly by 2.69% and 4.68% MIoU in 1-shot and 5-shot scenarios, respectively.

## 1. Introduction

In recent years, advancements in large-scale annotated datasets and deep neural networks (Chen et al., 2014; Long

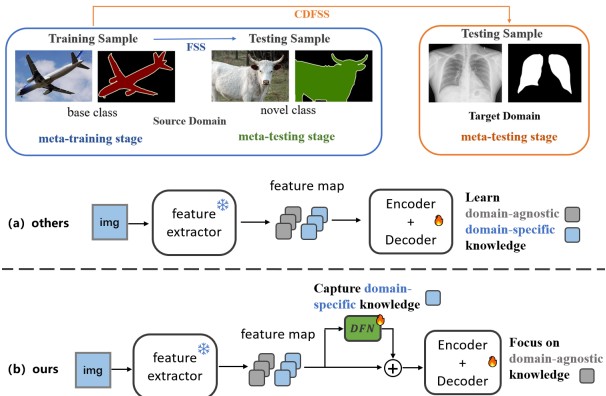

Figure 1: Cross-domain few-shot segmentation (CD-FSS) aims to transfer the source-domain-trained model to target domains for efficient learning with scarce data. By inserting adapters into the common network structure (e.g., HSNet) for CD-FSS, we find an insight not explored in previous works: adapter naturally serves as a domain information decoupler based on its structure instead of training losses, which *"grabs"* domain information from the encoder+decoder structure and encourages the model to learn domain-agnostic information on the source domain.

et al., 2015; Zhao et al., 2017; Yuan et al., 2020) have driven the rapid progress of large vision models (Dosovitskiy et al., 2020; Kirillov et al., 2023; Zhang et al., 2024), resulting in impressive segmentation task outcomes. However, when applied to downstream tasks, these models face significant challenges when there is a substantial distributional difference between upstream pretraining data and downstream data, where collecting downstream data may be difficult. To address this issue, the Cross Domain Few-Shot Segmentation (CD-FSS) task (Lei et al., 2022) has been introduced (see Figure 1 top). CD-FSS involves pre-train a model on a source-domain dataset and then adapting it to generate pixel-level predictions for unseen categories in target-domain datasets with only a few annotated samples, which still remains challenging.

There are majorly two challenges in the CD-FSS task: (1) a huge domain gap between source and target domains, making it difficult for the generalization from the source dataset to the target dataset; and (2) extremely limited target domain

[1]School of Computer Science and Technology, Huazhong University of Science and Technology [2]Peking University. Correspondence to: ✉Yixiong Zou <yixiongz@hust.edu.cn>.

*Proceedings of the 42^{nd} International Conference on Machine Learning*, Vancouver, Canada. PMLR 267, 2025. Copyright 2025 by the author(s).

data, making it challenging for the model to adapt to the distribution of the novel domain. To address the second challenge, currently, a group of methods based on adapters have been proposed (Houlsby et al., 2019; Mahabadi et al., 2021; Hu et al., 2021), which fixes the backbone network and only finetune the extra appended structures. In this paper, we revisit the adapter-based methods, and discover an intriguing insight not explored in prior works: adapters not only help the fine-tuning on downstream tasks but also **naturally serves as a domain information decoupler**. This finding indicates that the adapter can address two challenges simultaneously: (1) by decoupling the source domain information into a domain-agnostic part, which aids in generalizing from the source to the target domain, and (2) by parameter-efficient fine-tuning to adapt to downstream data.

In this paper, we first delve into this phenomenon for an interpretation. We first conduct experiments to verify which factors determine the adapter serving as a decoupler, including the adapter's insertion position and structure. We find this phenomenon holds only when adapters are inserted into deeper layers of the backbone network with scratch training and residual connections, without applying any domain-decoupling loss. This indicates **the structural design could lead to the inherent capability of domain-information decoupling**, which inspires us to design a structure-based domain decoupler, instead of the loss-based decoupler adopted by current works (Tzeng et al., 2015; Motiian et al., 2017; Kang et al., 2019; Lu et al., 2022).

Based on these findings and interpretations, we propose the Domain Feature Navigator (DFN, Fig. 1), a structure-based domain decoupler built on adapters with specific structures and positions. In the source-domain phase, DFN absorbs domain-specific knowledge without any domain-decoupling losses, directing the model's attention toward acquiring domain-agnostic information. Then, during the target-domain phase, we fine-tune the DFN to capture target-specific features. The fusion of domain-specific features with the model's domain-agnostic features serves to align the feature spaces for each domain.

However, the implicit absorption of domain information could potentially lead to excessive overfitting of source-domain samples, as the learning of domain information can also be understood as a kind of overfitting (to the source domain), but there are no labels to guide the magnitude of overfitting like other loss-based decouplers. To further address this problem, we introduce SAM-SVN to constrain the DFN from excessive overfitting in source-domain training. Specifically, we tailor the sharpness-aware minimization (Foret et al., 2020) to apply it to the singular value matrix of the DFN. By doing so, excessive overfitting is avoided but the absorption of domain information is maintained.

To sum up, our primary contributions are as follows:

- To the best of our knowledge, we are the first to discover the phenomenon that the adapter naturally serves as a decoupler, which we then delve into for an interpretation.
- Building upon this finding and interpretation, we propose the DFN to decouple source domain information into domain-agnostic knowledge and domain-specific one, solely based on the adapter's structure and position.
- We propose the SAM-SVN to avoid the potential excessive overfitting introduced by source-domain training of the DFN, while maintaining the DFN's absorption of domain information.
- Extensive experiments show the effectiveness of our work on four different CD-FSS scenarios. Our model significantly outperforms the state-of-the-art method.

## 2. Adapter Naturally Serves as Decoupler: Phenomenon and Interpretation

In this section, we conduct experiments to demonstrate that an adapter can naturally act as a decoupler. Furthermore, we examine which type of adapter is suitable for this role and investigate the underlying reasons. Following the standard CD-FSS setting, we use Pascal (Shaban et al., 2017) as the source dataset and the other four datasets as target datasets.

### 2.1. Adapter Decouples Domain Information

The network structure studied in this paper is shown in Fig.2 (top), which consists of a backbone network, an encoder, and a decoder. We choose this structure because it is widely recognized as versatile architecture (Min et al., 2021). We first attempt to attach a simple adapter (implemented as a $1 \times 1$ convolution) to the backbone with the residual connection, then train them jointly in the source domain.

Then, given ResNet50 (He et al., 2016) as the backbone, we measure the domain similarity by the CKA[1] (Kornblith et al., 2019; Zou et al., 2022; Tong et al., 2024b; Zou et al., 2024c) similarity (lower values indicate more domain-specific information) based on the backbone's stage-4 output before and after attaching the adapter (Fig.2). As shown in Table 1 (row1,2), after training with an adapter attached, the CKA is lower, which means the adapter captures domain-specific information, therefore decreasing the domain similarity.

Subsequently, we study the change that the adapter brings to the output of the encoder (which is different from the backbone network). We also measure the domain similarity by the CKA similarity through the encoder's output with and without the adapter attached to the backbone network (BKB). As shown in Table 1 (row 3,4), the CKA is higher after the adapter is attached, which implies that the encoder focuses more on the domain-agnostic information.

Without the adapter (Fig. 2 top), the encoder needs to si-

---

[1]See the Appendix B for detailed formulations.

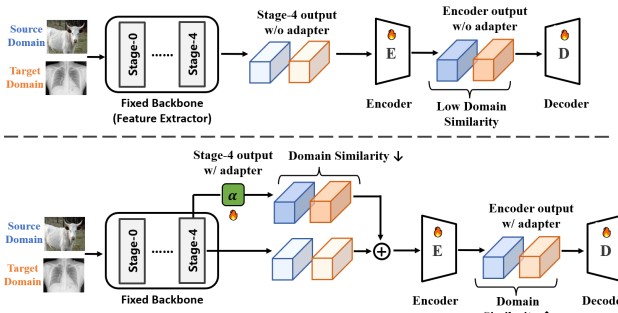

Figure 2: Network structure studied in this paper, we analyze stage-4 and encoder outputs to study the absorbed domain information by measuring the domain similarity.

| datasets | FSS-1000 | Deepglobe | ISIC | ChestX |
|---|---|---|---|---|
| stage-4 (w/o adapter) | 0.5371 | 0.4147 | 0.5266 | 0.4865 |
| stage-4 (w/ adapter) | 0.4605↓ | 0.3426↓ | 0.4459↓ | 0.3875↓ |
| encoder (w/o BKB adapter) | 0.0709 | 0.0498 | 0.0678 | 0.0554 |
| encoder (w/ BKB adapter) | 0.0733↑ | 0.0679↑ | 0.0724↑ | 0.0626↑ |

Table 1: Domain similarities for the output of stage-4 or encoder between Pascal and target datasets with and without adapter (Fig. 2). The adapter captures domain-specific information, guiding the encoder and decoder to learn domain-invariant knowledge. BKB: backbone network.

multaneously capture both domain-specific and domain-agnostic information extracted by the feature extractor. After attaching the adapter (Fig. 2 bottom), the adapter captures domain-specific information, therefore decreasing the BKB's domain similarity. Consequently, the encoder parameters could focus less on the domain-specific information and pay more attention to the domain-agnostic one, increasing the domain similarity. In other words, the adapter decouples the domain information from the original model.

## 2.2. Why adapter can serve as a decoupler?

The above phenomenon inspires us to ask: *Can all types of adapters decouple features, and why are adapters able to decouple features?* In this section, we delve into this phenomenon by exploring two factors that determine whether an adapter can be a decoupler: position and structure.

### 2.2.1. POSITION

We categorize the insertion positions of adapters into three types as shown in Fig 3: (1) shallow layers of the fixed backbone; (2) deep layers of the fixed backbone; (3) between learnable encoder and decoder. We use CKA to assess if position affects an adapter's decoupling ability.

Following Table 1, if the CKA of the backbone output decreases and that of the encoder output increases, we can recognize the attached adapter as a decoupler. As shown in Table 2, the adapter can serve as a decoupler only when the adapter is inserted into the deeper layers of the backbone.

Comparing the positions (2) and (3), the distinction is that

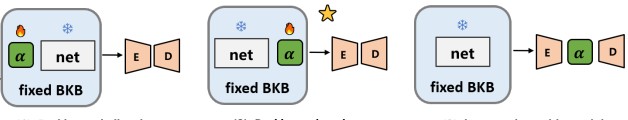

Figure 3: Three different positions for adapters.

| Position | baseline(w/o adapter) | | BKB shallower | | BKB deeper | | between enc-dec | |
|---|---|---|---|---|---|---|---|---|
| | BKB | encoder | BKB | encoder | BKB | encoder | BKB | encoder |
| FSS-1000 | 0.5371 | 0.0709 | 0.4679↓ | 0.0737↑ | 0.4788↓ | 0.0733↑ | 0.5371- | 0.0695↓ |
| Deepglobe | 0.4147 | 0.0498 | 0.3736↓ | 0.0452↓ | 0.3687↓ | 0.0679↑ | 0.4147- | 0.0469↓ |
| ISIC | 0.5266 | 0.0678 | 0.5028↓ | 0.0633↓ | 0.4653↓ | 0.0724↑ | 0.5266- | 0.0658↓ |
| Chest X-ray | 0.4865 | 0.0554 | 0.4639↓ | 0.0532↓ | 0.4390↓ | 0.0626↑ | 0.4865- | 0.0529↓ |

Table 2: Verify the impact of different insertion points of the adapter on its decoupling ability by domain similarity.

(2) is positioned between the fixed, pretrained backbone and subsequent learnable modules. The pretrained backbone is fixed on the source domain, while the adapter is trained from scratch. Such a difference guides the backbone network to extract general features and leads the adapter to learn more from the source domain to capture domain information.

Comparing insertion methods (1) and (2), the distinction is that (2) is located in the deeper layers of the fixed backbone, whereas (1) is positioned in the shallower layers. As is well known, the deeper the neural network is, the more semantic information its features can encompass. For cross-domain tasks, the knowledge learned by deeper network layers tends to be more domain-specific. To verify it, we first visualized feature maps with different examples using insertion method (2), and the results are shown in Fig.4. For each example, the adapter outputs more complex features to focus on the objects' profile, such as the eagle's wings, fish tail and fins, and the clock face of the clock tower, which means that inserting the adapter into deeper layers enables it to capture features that are more semantic and complex.

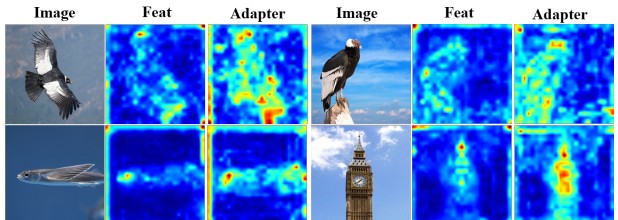

Figure 4: The visualization results of the feature maps with and without the adapter attached to the backbone network.

Furthermore, as shown in Table 3, we evaluate the CKA similarity between features extracted at deep layers and that of the first convolution layer (conv1) in the source domain following (Zou et al., 2022). The higher the CKA similarity, the simpler the features are. The row results indicate that the deeper the layer is, the lower the CKA would be. This shows high-level features are more complex, which means it could be more overfitting to data, e.g., domain information. The second row's adapters are attached to the convolutional layer

| backbone | conv1 | conv3_4 | conv4_6 | conv5_3 |
|---|---|---|---|---|
| conv output | 1.0 | 0.8768 | 0.8496 | 0.8164 |
| adapter output | 1.0 | 0.8707 | 0.8364 | 0.7936 |

Table 3: Validate the complexity of different layers' features by the similarity with the first-layer feature.

in the first row. Comparing these two rows, we can see that the CKA is lower after the feature passes through the adapter, which means the adapters' features are more complex and could capture more domain-specific knowledge.

**Conclusion for position:** (1) The disparity between the backbone network (pretrained, fixed) and the adapter (learnable, from scratch), enables the adapter to acquire specific features during training. (2) The semantics become richer in deeper layers, for cross-domain tasks, the knowledge learned by deeper network layers tends to be more domain-specific. These two factors enable the adapter to act as a decoupler when it is inserted into the deeper layers of the fixed, pretrained backbone network.

### 2.2.2. STRUCTURE

As shown in Figure 5, we explore two design structures, conventional and LoRA, and two connection structures, serial (ser) and residual (res). We adopt CKA to assess if structure affects an adapter's decoupling ability as shown in Table 4.

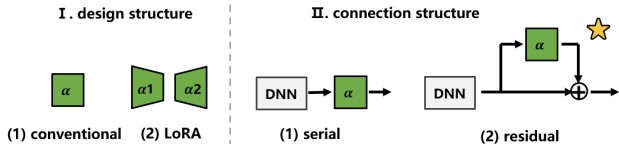

Figure 5: Two different design and connection structures.

| structure | w/o adapter | | conventional+res | | LoRA+res | | conventional+ser | |
|---|---|---|---|---|---|---|---|---|
| | stage-4 | enc | adapter | enc | adapter | enc | adapter | enc |
| FSS-1000 | 0.5371 | 0.0709 | 0.4605↓ | 0.0733↑ | 0.4529↓ | 0.0752↑ | 0.4906↓ | 0.0623↓ |
| Deepglobe | 0.4147 | 0.0498 | 0.3426↓ | 0.0679↑ | 0.3503↓ | 0.0628↑ | 0.3932↓ | 0.0415↓ |
| ISIC | 0.5266 | 0.0678 | 0.4459↓ | 0.0724↑ | 0.4537↓ | 0.0705↑ | 0.4752↓ | 0.0653↓ |
| ChestX | 0.4865 | 0.0554 | 0.3875↓ | 0.0626↑ | 0.4021↓ | 0.0597↑ | 0.4210↓ | 0.0506↓ |

Table 4: Assess the impact of different adapter structures on their decoupling capabilities by domain similarity.

Both design structures lead to reduced CKA for features passing through the adapter and increased CKA for features through the encoder, indicating that the adapter learned domain-specific knowledge while the encoder focused on domain-agnostic information. However, changing the connection from residual to serial leads to a decrease in the encoder output feature's CKA, making it more domain-specific. This suggests that an adapter's ability to decouple domain information is independent of its design structure and solely related to its connection structure.

We hold that this is because the residual connection **explicitly** separates and integrates the backbone feature (containing general information) and the adapter feature (contain-

ing domain information) to be transmitted to subsequent modules. In contrast, the serial connection allows only the adapter's domain-specific features to be transmitted to the subsequent learnable modules.

**Conclusion for structure:** the residual connection explicitly partitions the domain-specific features learned by the adapter from the general features extracted by the backbone, before jointly feeding them into subsequent learnable structures, thereby promoting subsequent modules to focus on invariant information. Therefore, the residual connection is crucial for an adapter to act as a decoupler, while structural differences do not affect its decoupling capability.

**Discussion.** Inserting the adapter with residual connections deep into the pre-trained and fixed backbone, and before subsequent learnable modules, enables it to act as a domain knowledge decoupler. The above findings and interpretations indicate the structural design of the feature extractor can lead to the inherent capability of domain-information decoupling, which is different from the decoupling achieved by applying domain losses adopted by current works (Motiian et al., 2017; Kang et al., 2019; Lu et al., 2022). This inspires us to design a structure-based decoupler further to make better use of such inherent capability.

| Method | FSS | Deepglobe | ISIC | ChestX | Method | loss fluc. |
|---|---|---|---|---|---|---|
| Baseline | 1.43 | 2.18 | 2.22 | 1.57 | Baseline | 0.398 |
| Baseline + adapter | 1.76 | 2.33 | 3.12 | 2.03 | Baseline+ad. | 0.521 |

Table 5: Sharpness of loss landscapes measured by the loss fluctuations, where higher sharpness means a higher tendency of overfitting. **Left:** Perturbation by parameter initialization. **Right:** Perturbation by Gaussian noises.

However, solely the structure-based decoupler can also lead to problems: the implicit absorption of domain information has no guarantee for its correctness. For example, the adapter generally absorbs deeper and more complex patterns that are more likely to be domain information, but it also enables the model to overfit specific training samples by such complex patterns, which is harmful for downstream generalization and adaptation. To validate this risk, we are inspired by (Foret et al., 2020; Zou et al., 2024a) to test the sharpness of the model's loss landscapes to verify its tendency of overfitting, as shown in Fig. 6 (Flatten loss landscape)[2]. The perturbation is added in two ways: different initialization of the model parameters in Tab. 5 (left) and random Gaussian noise to the model parameters in Tab. 5 (right). We can see both kinds of perturbation lead to increased sharpness of loss landscapes, indicating a higher tendency to overfit training samples. Since absorbing domain information can also be understood as a kind of overfitting (to the source domain), there are no specific criteria to determine whether the overfitting is just for the domain instead of samples,

---

[2]See the Appendix D for a detailed explanation of sharpness.

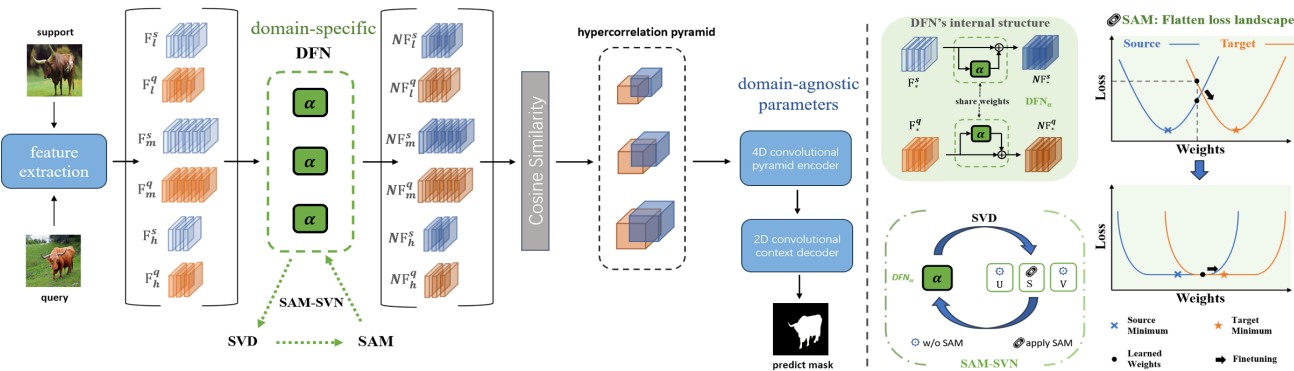

Figure 6: Overview of our method in a 1-shot example. After obtaining the pyramid features of support and query images, DFN is introduced to accumulate the domain-specific knowledge then encouraging the model to learn domain-agnostic parameters. Moreover, we propose SAM-SVN to eliminate potential excessive overfitting introduced by source domain training on DFN. The internal structure of DFN and SVN-SAM is highlighted in green.

unlike the loss-based decoupler where the model is guided by the domain labels. Therefore, during source-domain training, we need to constrain the adapter from excessive overfitting (to enhance stability during target-domain fine-tuning), while also preserving its decoupling capability (to address the domain gap).

## 3. Method

Building upon the insight that adapters can naturally serve as decouplers, we propose DFN, a variant of adapters, to simultaneously handle the domain gap and data scarcity problem for CD-FSS. As in Fig 6, during the source-domain training, the DFN is jointly trained with the base model, we connect the DFN to low, middle, and high-level features separately to ensure semantic consistency. Moreover, we design SAM-SVN to prevent the risk of excessive overfitting on DFN introduced by source-domain training that hinders downstream generalization and fine-tuning.

### 3.1. Problem Definition

Given a source domain $D_s = (P(X_s), Y_s)$ and a target domain $D_t = (P(X_t), Y_t)$, where $P(X)$ represents input data distribution and $Y$ represents label space. Each class $c_s \in D_s$ has sufficient labels while $c_t \in D_t$ has only limited labels. Notably, the source domain $D_s$ and target domain $D_t$ exhibit distinct input data distribution, with their respective label spaces having no intersection, i.e., $P(X_s) \neq P(X_t)$, $Y_s \cap Y_t = \emptyset$. The model will be trained on the dataset from the $D_s$, then applied to segment novel classes in the $D_t$.

### 3.2. DFN: Domain Feature Navigator

The role of DFN is to absorb domain information to encourage the model to acquire domain-agnostic information on the source domain, and adapt the model to target domains. Since in section 2 we have concluded under what condition an adapter can be a decoupler, we term the adapter satisfying

such conditions as the proposed DFN.

Therefore, we use a feature extractor to extract features and attach DFN to the features by residual connection (see Figure.6 green box for DFN's internal structure). Concretely, let $\mathcal{N}_\alpha$ denote the module DFN with parameters $\alpha \in \mathbb{R}^{C \times C \times 1 \times 1}$, i.e., $\mathcal{N}_\alpha$ is implemented as a convolutional operation with $1 \times 1$ kernels. It maintains the same number of input and output channels. $f_\phi$ denote the feature extractor, a sequence of $L$ pairs of intermediate feature maps $\{(F_l^q, F_l^s)\}_{l=1}^L$ is produced by $f_\phi$. We then mask each support feature map $F_l^s \in \mathbb{R}^{C_l \times H_l \times W_l}$ using the support mask $M^s \in \{0,1\}^{H \times W}$ to discard irrelevant activations for reliable mask prediction:

$$F_l^s = f_{\phi_l}(x), \quad \hat{F}_l^s = F_l^s \odot \zeta_l(M^s) \tag{1}$$

where $x \in \mathbb{R}^{C \times H \times W}$ is the input tensor, $\odot$ is Hadamard product, and $\zeta_l(*)$ is a function that bilinearly interpolates input tensor to the spatial size of the feature map $F_l^s$ at layer $l$ followed by expansion along channel dimension such that $\zeta_l : \mathbb{R}^{H \times W} \to \mathbb{R}^{C_l \times H_l \times W_l}$.

The final outputs of navigated feature maps are:

$$\{NF_l^s\}_{l=1}^L = \{\hat{F}_l^s\}_{l=1}^L + \mathcal{N}_\alpha(\{\hat{F}_l^s\}_{l=1}^L) \tag{2}$$
$$\{NF_l^q\}_{l=1}^L = \{F_l^q\}_{l=1}^L + \mathcal{N}_\alpha(\{F_l^q\}_{l=1}^L) \tag{3}$$

where $NF_l^*$ is the feature that has been aligned to the target domain feature space after passing through the $\mathcal{N}_\alpha$.

For the subsequent hypercorrection pyramid construction, a pair of navigated query features and navigated masked support features at each layer forms a 4D correlation tensor $C_l \in \mathbb{R}^{H_l \times W_l \times H_l \times W_l}$ using cosine similarity:

$$C_l(m,n) = ReLU\left(\frac{NF_l^q(m) \cdot NF_l^s(n)}{\|NF_l^q(m)\|\|NF_l^s(n)\|}\right) \tag{4}$$

where $m, n$ denote 2D spatial positions of navigated feature maps (i.e. have been decoupled by DFN) $NF_l^q$ and $NF_l^s$

respectively. Then, the $C_l(m,n)$ is fed into the 4D convolutional pyramid encoder and the 2D convolutional decoder to obtain segmentation results, as shown in Figure 6.

### 3.3. SAM-SVN: Perturbing only the Singular Value of the Domain Feature Navigator

The absorption of domain information by solely the structure-based adapter could lead to problems: if DFN learns overly complex patterns, it may overfit source samples rather than the source domain, represented as increased sharpness against perturbations, as verified in Table 5.

To handle this problem, we are inspired by the Sharpness-Aware Minimization (SAM) (Foret et al., 2020), which was proposed to reduce the sharpness of minima and has been shown to resist overfitting in various settings (Kaddour et al., 2022; Mueller et al., 2023). However, simply resisting the sharpness can also limit the absorption of domain information, since the learning of domain information can also be understood as a kind of overfitting (to the source domain).

To overcome this issue, inspired by SAM-ON (Mueller et al., 2023), which perturbs only the normalization layers, and building on BSP (Chen et al., 2019), which proposed that singular values control the importance of different representations during transfer, we developed a tailored enhancement for the DFN, called SAM-SVN (Fig.6, green box for SAM-SVN). In essence, we first perform singular value decomposition on the DFN, then apply SAM to the singular value matrix. Since only the most overfitting-sensitive parameters (singular values) are constrained, the remaining parameters are still capable of absorbing domain information.

In detail, for a DFN $\mathcal{N}_\alpha$ with $C_i$ input channels, $C_o$ output channels, and a kernel size of $K \times K$, we first fold its weight tensor $\alpha \in \mathbb{R}^{C_o \times C_i \times K \times K}$ into a matrix $\alpha' \in \mathbb{R}^{C_o \times C_i K^2}$, then decompose the obtained matrix by applying SVD with full-rank in subspaces (rank $R = min(C_o, C_i K^2)$ ). Thus,

$$\alpha' = USV^T \qquad (5)$$

where $U \in \mathbb{R}^{C_o \times R}$, $S \in \mathbb{R}^{R \times R}$, and $V^T \in \mathbb{R}^{R \times C_i K^2}$. In our setting, we have $K = 1$ and input channels as same as output channels (i.e., $C_i = C_o$). The obtained pair of matrices $V^T$ and $U$ construct two new convolution layers, and $S$ is a diagonal matrix with singular values on the diagonal.

Let's redirect our attention towards the SAM. We consider a neural network $f_w$ attached with the DFN $\mathcal{N}_\alpha$ as our model $\mathcal{M}_\varphi$ ($\varphi = \{w, \alpha\}$). The training sample $(x, y)$ consists of input-output pairs are drawn from the source domain $D_s$. Following HSNet (Min et al., 2021), we employ the standard Binary Cross-Entropy (BCE) loss, denoted as $L$, to train the model $\mathcal{M}$, aiming to minimize $L$ over the training set. Conventional SGD-like optimization methods minimize $L$ by stochastic gradient descent $\nabla$. SAM aims at additionally minimizing the worst-case sharpness of the training loss in

a neighborhood defined by an $\ell_p$ ball around matrix $\varphi$.

In practice, SAM uses $p = 2$ and approximates the inner maximization by a single gradient step, yielding:

$$\epsilon = \rho \nabla L(S) / \|\nabla L(S)\|_2 \qquad (6)$$

and requiring an additional forward-backward pass compared to SGD. The gradient is then re-evaluated at the perturbed point $\alpha + \epsilon$, giving the actual weight update:

$$\hat{\alpha} = U(S + \epsilon)V^T \qquad (7)$$
$$w \leftarrow w - \beta \nabla L(w, \hat{\alpha}) \qquad (8)$$
$$\alpha \leftarrow \alpha - \beta \nabla L(w, \hat{\alpha}) \qquad (9)$$

In this endeavor, SAM is applied to the singular value matrix $S$ obtained through the SVD of DFN. The prediction BCE loss $L$ is calculated using the updated $\{w, \alpha\}$.

During the target-domain finetuning stage, we fine-tune the DFN to learn domain-specific features, the fusion of domain-specific features with the model's domain-agnostic features serves to align the feature spaces for each domain.

## 4. Experiments

**Datasets.** We adopt the benchmark proposed by PATNet (Lei et al., 2022) and follow the same data preprocessing procedures. We employ PASCAL (Shaban et al., 2017) , which is an extended version of PASCAL VOC 2012 (Everingham et al., 2010), as our source-domain dataset for training. We regard FSS-1000 (Li et al., 2020), Deepglobe (Demir et al., 2018), ISIC2018 (Codella et al., 2019; Tschandl et al., 2018), and Chest X-ray (Candemir et al., 2013; Jaeger et al., 2013) as target domains for evaluation.

**Implementation Details.** We employ ResNet-50 (He et al., 2016) pre-trained on ImageNet (Russakovsky et al., 2015) as our backbone, with its weights frozen during training, following HSNet (Min et al., 2021). To optimize memory usage and speed up training, we set the spatial sizes of both support and query images to 400 × 400. The model is trained using the Adam (Min et al., 2021) optimizer with a learning rate of 1e-3. The hyperparameter $\rho$ in SAM is set to 0.5. We also implemented our method in the transformer architecture following FPTrans (Zhang et al., 2022), employing ViT (Dosovitskiy et al., 2020) as the backbone. The fine-tuning of DFN is performed using the Adam optimizer, with learning rates set at 1e-3 for FSS-1000, 5e-1 for Deepglobe, 5e-3 for ISIC and Chest X-ray. Each task undergoes a total of 50 iterations.

### 4.1. Comparison with State-of-the-arts

In Table 6, we compare our method with several state-of-the-art few-shot segmentation methods on benchmark proposed by PATNet (Lei et al., 2022). The results reveal that the performance of existing few-shot semantic segmentation

| Method | Mark | Backbone | FSS-1000 | | Deepglobe | | ISIC | | Chest X-ray | | Average | |
|---|---|---|---|---|---|---|---|---|---|---|---|---|
| | | | 1-shot | 5-shot | 1-shot | 5-shot | 1-shot | 5-shot | 1-shot | 5-shot | 1-shot | 5-shot |
| CaNet (Zhang et al., 2019) | CVPR-19 | Res-50 | 70.67 | 72.03 | 22.32 | 23.07 | 25.16 | 28.22 | 28.35 | 28.62 | 36.63 | 37.99 |
| PANet (Wang et al., 2019) | ECCV-20 | Res-50 | 69.15 | 71.68 | 36.55 | 45.43 | 25.29 | 33.99 | 57.75 | 69.31 | 47.19 | 55.10 |
| RPMMs (Yang et al., 2020) | ECCV-20 | Res-50 | 65.12 | 67.06 | 12.99 | 13.47 | 18.02 | 20.04 | 30.11 | 30.82 | 31.56 | 32.85 |
| PFENet (Tian et al., 2020) | TPAMI-20 | Res-50 | 70.87 | 70.52 | 16.88 | 18.01 | 23.50 | 23.83 | 27.22 | 27.57 | 34.62 | 34.98 |
| RePRI (Boudiaf et al., 2021) | CVPR-21 | Res-50 | 70.96 | 74.23 | 25.03 | 27.41 | 23.27 | 26.23 | 65.08 | 65.48 | 46.09 | 48.34 |
| HSNet (Min et al., 2021) | ICCV-21 | Res-50 | 77.53 | 80.99 | 29.65 | 35.08 | 31.20 | 35.10 | 51.88 | 54.36 | 47.57 | 51.38 |
| PATNet (Lei et al., 2022) | ECCV-22 | Res-50 | 78.59 | 81.23 | 37.89 | 42.97 | 41.16 | 53.58 | 66.61 | 70.20 | 56.06 | 61.99 |
| PATNet (Lei et al., 2022) | ECCV-22 | ViT-base | 72.03 | - | 22.37 | - | 44.25 | - | 76.43 | - | 53.77 | - |
| RestNet (Huang et al., 2023) | BMVC-23 | Res-50 | 81.53 | 84.89 | 22.70 | 29.90 | 42.25 | 51.10 | 70.43 | 73.69 | 54.22 | 59.89 |
| PerSAM (Zhang et al., 2024) | ICLR-24 | ViT-base | 60.92 | 66.53 | 36.08 | 40.65 | 23.27 | 25.33 | 29.95 | 30.05 | 37.56 | 40.64 |
| ABCDFSS (Herzog, 2024) | CVPR-24 | Res-50 | 74.60 | 76.20 | 42.60 | 45.70 | 45.70 | 53.30 | 79.80 | 81.40 | 60.67 | 64.97 |
| APSeg (He et al., 2024) | CVPR-24 | ViT-base | 79.71 | 81.90 | 35.94 | 39.98 | 45.43 | 53.98 | 84.10 | 84.50 | 61.30 | 65.09 |
| APM (Tong et al., 2024b) | NeurIPS-24 | Res-50 | 79.29 | 81.83 | 40.86 | 44.92 | 41.71 | 51.76 | 78.25 | 82.81 | 60.03 | 65.18 |
| **DFN (Ours)** | Ours | Res-50 | 80.73 | **85.80** | **45.66** | **47.98** | 36.30 | 51.13 | **85.21** | **90.34** | 61.98 | 68.81 |
| **DFN (Ours)** | Ours | ViT-base | **82.97** | 85.72 | 39.45 | 47.67 | **50.36** | **58.53** | 83.18 | 87.14 | **63.99** | **69.77** |

Table 6: MIoU of 1-shot and 5-shot results on the CD-FSS benchmark. The best and second-best results are in bold and underlined, respectively. In the appendix E/F, we conduct experiments on hyper-parameters and computational complexity.

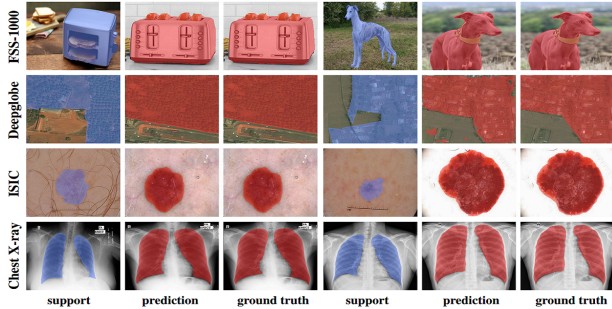

Figure 7: Qualitative results of our model for 1-shot setting.

methods degrades significantly under domain shifts. Table 6 indicates a significant advancement in cross-domain semantic segmentation for both 1-shot and 5-shot tasks using our proposed approach. Specifically, we surpass the performance of the state-of-the-art APSeg by 2.69% (1-shot Ave.) and 4.68% (5-shot Ave.), respectively. Furthermore, we showcase qualitative results of the proposed method in 1-way 1-shot segmentation, as depicted in Figure 7.

### 4.2. Ablation Study

**Effectiveness of each component.** We analyze each proposed component on the 1-shot and 5-shot settings to validate the effectiveness of each design, including DFN and SAM-SVN (SAM applied on the Singular Value Matrix of Navigator, i.e., SAM and SVD shown in the table). The results, presented in Table 7 (left), indicate that introducing DFN improved average MIoU by 12.32% for 1-shot and 10.21% for 5-shot, while adding SAM-SVN further increased it by 2.09% and 2.22%, respectively. These results demonstrate that each design in our approach significantly contributes to performance gains. (see appendix for ViT)

**SAM perturbation targets.** We explored the impact of ap-

| DFN | SAM | SVD | 1-shot | 5-shot |
|---|---|---|---|---|
| | | | 47.57 | 51.38 |
| ✓ | | | 59.89 | 66.59 |
| ✓ | ✓ | | 60.65 | 67.74 |
| ✓ | ✓ | ✓ | **61.98** | **68.81** |

| apply SAM | 1-shot | 5-shot |
|---|---|---|
| Enc.+Dec.+DFN | 60.04 | 66.98 |
| DFN | 60.65 | 67.74 |
| SVN | **61.98** | **68.81** |

Table 7: Ablation study on various designs (left) and apply SAM on different modules (right). Enc: Encoder, Dec: Decoder, SVN: Singular Value matrix of the DFN.

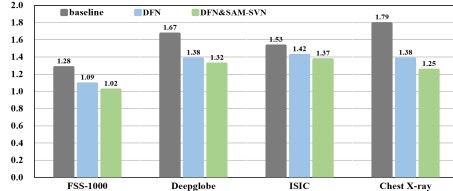

Figure 8: Our designs reduce the feature's domain distance, enhancing the model's learning of agnostic knowledge.

plying SAM to different modules, detailed in Tab 7 (right). SAM was applied to DFN, Encoder, and Decoder collectively, exclusively to DFN, and on SVN. Results highlight the effectiveness of SAM on SVN. SAM prevents DFN from overfitting to the source domain, aiding fine-tuning, but it may affect domain-specific knowledge capture. SAM-SVN strikes a balance, preventing DFN from overfitting to the source domain without compromising its ability to capture domain-specific information during training.

**Improvement in model generalization.** We evaluate our method's impact on model output using Maximum Mean Discrepancy (MMD[3]) (Gretton et al., 2012) to measure domain distance of target datasets with respect to the PASCAL. A lower MMD indicates reduced domain distance, meaning the model produces more agnostic features. Figure 8 shows

---

[3]The CKA measure is in the appendix showing similar results.

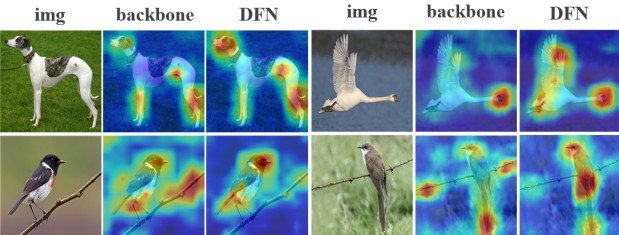

Figure 9: Feature map visualization: (row 1) DFN guides model's attention to more distinctive features. (row 2) DFN guides the model to perform more accurate segmentation.

that attaching DFN to the backbone reduces the MMD, and further it decreases after applying SAM-SVN. This suggests the model has acquired more agnostic representations, reducing the distance between domains.

### 4.3. DFN Guiding Model Attention

During source-domain training, the DFN absorbs domain-specific knowledge, directing the model's attention toward acquiring domain-agnostic insights. The impact of DFN is demonstrated through feature map visualization (Figure 9). In the first row, without DFN, the model focuses on general features such as the dog's head, limbs, and tail, and the swan's head. After passing through the DFN, the model is guided to pay attention to more distinctive and specific features, such as the dog's head and tail, and the swan's wing. In the second row, the model focuses on various objects, namely the bird and the tree branch. After passing through the DFN, the model is guided to pay attention specifically to the object we want to segment (the bird).

### 4.4. SAM-SVN Reduces Sharpness and Improve Robustness to Domain Shifts

The motivation behind the SAM-SVN is primarily twofold: 1) to limit the DFN's acquisition of excessive source-specific knowledge; and 2) to avoid hindering the DFN's decoupling ability, as this might undermine the model's learning of invariant knowledge. The SAM-SVN can achieve both goals by reducing the sharpness of the training loss against DFN's parameter, because (1) it can prevent DFN from being vulnerable to domain shifts, and (2) it does not harm the minimization of the training loss.

To verify the reduced sharpness and improved robustness of domain shifts attributed to the SAM-SVN, as shown in Figure 10, we adopted style transfer methods to superimpose target-domain styles as domain shifts (horizontal axis). The lowered performance drop (vertical axis) validates the flattened loss landscapes and the enhanced robustness to domain shifts. We also measured the impact of SAM-SVN on model's performance stability, as shown in Table 8. By constraining DFN from absorbing sample-specific information (flatten DFN's loss landscape) without affecting its absorption of domain-specific information, SAM-SVN helps improve model stability (lower performance fluctuation).

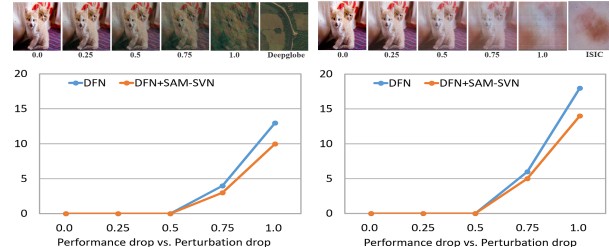

Figure 10: After adopting SAM-SVN, the model exhibits reduced sharpness and enhanced robustness to domain shifts.

| Method | FSS-1000 | Deepglobe | ISIC | ChestX |
|---|---|---|---|---|
| Only DFN | 1.76 | 2.33 | 3.12 | 2.03 |
| DFN + SAM | 1.25 | 2.92 | 2.85 | 1.89 |
| DFN + SAM-SVN | 0.94 | 1.53 | 1.68 | 1.18 |

Table 8: Train five best checkpoints on the same device and measure 1-shot performance fluctuations ($best - worst$).

## 5. Analysis of DFN Usage and Adapter Choice

**Impact of DFN's usage manner.** We examined different strategies for utilizing DFN: 1) using DFN in source domain training only, removing it in target domain; 2) using DFN during source training, but re-initializing its parameters and re-trained from scratch during target domain finetuning. Table 9 shows that even when DFN is removed in the target domain, it still improves performance over the baseline by guiding the model toward domain-agnostic representations. However, this remains suboptimal due to the lack of adaptation to target-specific information. Re-initializing and retraining DFN during target finetuning, performance varies with the initialization method, yet remains consistently strong across all variants. Our approach (i.e., DFN + SAM-SVN) is seen as using DFN to guide the model to focus on domain-agnostic knowledge, while SAM-SVN provides DFN with a reasonable initialization value beneficial for adapting to various domains.

| 1-shot Setting | FSS1000 | Deepglobe | ISIC | ChestX | Mean |
|---|---|---|---|---|---|
| Baseline | 77.53 | 29.65 | 31.20 | 51.88 | 47.57 |
| DFN (remove in target) | 78.16 | 38.21 | 34.12 | 76.92 | 56.85 |
| DFN (scratch, Kaiming) | 79.02 | 42.53 | 36.63 | 82.03 | 60.05 |
| DFN (scratch, Xavier) | 79.83 | 45.57 | 34.79 | 79.46 | 59.91 |
| DFN (scratch, Gaussian) | 78.97 | 38.75 | 33.82 | 78.59 | 57.53 |
| DFN (Ours) | **80.73** | **45.66** | **36.30** | **85.21** | **61.98** |

Table 9: Impact of DFN usage manner on performance.

**Impact of adapter choice on sharpness** We further measure the impact of different adapter choices on sharpness. Consistent with Figure 3 and Figure 5, we analyze the results based on adapter position and structure (res: residual, ser: serial, BKB: backbone, enc-dec: encoder-decoder). The results in Table 10 and Table 11 show that, from the perspective of sharpness, significant changes in loss fluctuations occur only when residual links are satisfied and the position is deep within the backbone (with the adapter structure not

| Position | Baseline | BKB shallower | BKB deeper | Between enc-dec |
|---|---|---|---|---|
| **Loss fluc.** | 0.398 | 0.402 | 0.521 | 0.405 |

Table 10: Impact of adapter position on loss fluctuations.

| Structure | Baseline | conventional+res | LoRA+res | conventional+ser |
|---|---|---|---|---|
| **Loss fluc.** | 0.398 | 0.521 | 0.533 | 0.399 |

Table 11: Impact of adapter structure on loss fluctuations.

being a determining factor), This indicates that the adapter has captured domain-specific related information, which is consistent with the conclusions drawn in section 2.

## 6. Theoretical Analysis for Adapter naturally Serves as Information Decoupler

The behavior of adapters as domain information decouplers can be analyzed through the Information Bottleneck (IB) theory. The IB objective is given by:

$$\mathcal{L}_{\text{IB}} = I(X; Z) - \beta I(Z; Y) \tag{10}$$

where $I(\cdot; \cdot)$ denotes mutual information, $X$ is the input, $Y$ is the final output label, $Z$ is the intermediate representation, and $\beta$ is the trade-off parameter.

Assume input $X$ can be decomposed into domain-invariant and domain-specific components $X = X_{\text{inv}} + X_{\text{spec}}$, then the IB objective becomes:

$$\mathcal{L}_{\text{IB}} = I(X_{\text{inv}} + X_{\text{spec}}; Z) - \beta I(Z; Y) \tag{11}$$

For encoder-decoder (ED) network with parameters $\theta_f$ and a adapter with parameters $\theta_g$ where the adapter has a limited capacity (i.e., $\theta_f \gg \theta_g$). Due to the lower capacity of the adapter (much smaller than the ED's parameters), it tends to absorb primarily the domain-specific signal, which is more informative for the current training objective. This leads to:

$$I(X_{\text{spec}}; Z_{\text{adapter}}) \gg I(X_{\text{inv}}; Z_{\text{adapter}}) \tag{12}$$

where $Z_{\text{adapter}}$ is the information processed or represented by the adapter. It promotes gradient flow differentiation. The forward function of the residual adapter structure is:

$$F(x) = f(x) + g(f(x)) \tag{13}$$

where $g(f(x))$ is the output of the adapter. The gradient w.r.t. the encoder-decoder parameters $\theta_f$ is:

$$\frac{\partial \mathcal{L}}{\partial \theta_f} = \frac{\partial \mathcal{L}}{\partial F(x)} \cdot \frac{\partial F(x)}{\partial f(x)} \cdot \frac{\partial f(x)}{\partial \theta_f} \tag{14}$$

Expanding the middle term:

$$\frac{\partial F(x)}{\partial f(x)} = \mathbf{I} + \frac{\partial g(f(x))}{\partial f(x)} \tag{15}$$

where $\mathbf{I}$ is the identity matrix (from the direct residual path), and the second term is the Jacobian matrix of the adapter function with respect to $f(x)$.

For the adapter parameters $\theta_g$, the gradient is:

$$\frac{\partial \mathcal{L}}{\partial \theta_g} = \frac{\partial \mathcal{L}}{\partial F(x)} \cdot \frac{\partial g(f(x))}{\partial \theta_g} \tag{16}$$

**Differentiated gradients:** Due to the adapter's selective absorption of domain-specific information and the residual adapter structure, gradient flow naturally separates network optimization into two complementary learning objectives. (Detailed derivation in appendix A)

## 7. Related Work

**Few-shot semantic segmentation** FSS (Tong et al., 2024a) aims to segment unseen semantic objects in query images with only a few annotated samples. OSLSM (Shaban et al., 2017) contributes to the first two-branch FSS model. Following this, PL (Dong & Xing, 2018) introduces a prototype learning paradigm, where predictions leverage the cosine similarity between pixels and prototypes. SG-One (Zhang et al., 2020) adopt the masked average pooling operation to enhance the extraction of support feature. HSNet (Min et al., 2021) employs efficient 4D convolutions on multi-level feature correlations, significantly enhancing performance and serving as the baseline for our work. However, these methods focus solely on segmenting novel classes within the same domain and struggle to generalize to unseen domains. Bridging the huge domain gap between source and target domains with limited labeled data, remains a challenge.

**Domain-invariant representation learning** Domain-invariant representation learning has been widely adopted to address domain shifts. (Tzeng et al., 2015) introduces domain confusion loss and soft label to render domains indistinguishable in feature space, promoting the learning of domain-invariant features. (Motiian et al., 2017) implements domain alignment and semantic alignment through CCSA loss. DIFEX(Lu et al., 2022) maximizes both invariant and domain-specific features through additional regularization terms to better leverage invariant characteristics. These methods aim to induce models to acquire invariant knowledge through restrictive mechanisms such as losses and regularization. Diverging from these loss-based approaches, our work leverages the inherent properties of adapters to propose a structure-based domain knowledge decoupler.

## 8. Conclusion

In this paper, we introduce a novel perspective: adapter naturally serve as domain information decoupler. Based on this, we propose the DFN to guide the model's attention towards acquiring domain-agnostic information. Additionally, we introduce the SAM-SVN to prevent the potential excessive overfitting on DFN introduced by source-domain training that hinders the acquisition of domain-specific knowledge during fine-tuning. Experimental results demonstrate the effectiveness of our approach in bridging domain gaps.

## Acknowledgements

This work is supported by the National Natural Science Foundation of China under grants 62206102; the National Key Research and Development Program of China under grant 2024YFC3307900; the National Natural Science Foundation of China under grants 62436003, 62376103 and 62302184; the National Natural Science Foundation of China under grants 62402015; the Postdocotoral Fellowship Program of CPSF under grants GZB20230024; the China Postdoctoral Science Foundation under grant 2024M750100; Major Science and Technology Project of Hubei Province under grant 2024BAA008; Hubei Science and Technology Talent Service Project under grant 2024DJC078; and Ant Group through CCF-Ant Research Fund. The computation is completed in the HPC Platform of Huazhong University of Science and Technology.

## Impact Statement

Based on our novel perspective that adapters naturally serve as decouplers, we propose a structure-based decoupling method to address the CD-FSS problem. This approach can also be applied in other areas such as few-shot learning and domain adaptation, since the widespread use of adapters in various fields. The limitation of this work is its oversight of many-shot scenarios, which have significant practical value in real-world applications. Nonetheless, our method is adaptable and can be further optimized to support many-shot scenarios in the future.

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

# Appendix for Adapter Naturally Serves as Decoupler for Cross-Domain Few-Shot Semantic Segmentation

## A. Detailed Derivation of Adapter Naturally Serves as Information Decoupler

Let the features $f(x)$ from the ED be conceptually decomposable into domain-invariant $f_{\text{inv}}(x)$ and domain-specific $f_{\text{spec}}(x)$ components:

$$f(x) = f_{\text{inv}}(x) + f_{\text{spec}}(x) \tag{17}$$

We assume the existence of orthogonal projection operators $P_{\text{inv}}$ and $P_{\text{spec}}$ onto conceptual domain-invariant and domain-specific subspaces, respectively, such that $f_{\text{inv}}(x) \approx P_{\text{inv}} f(x)$ and $f_{\text{spec}}(x) \approx P_{\text{spec}} f(x)$, with $P_{\text{inv}} + P_{\text{spec}} \approx \mathbf{I}$ and $P_{\text{inv}} P_{\text{spec}} \approx 0$.

**Step 1:** Decompose the gradient of the ED's features $f(x)$ w.r.t. its parameters $\theta_f$ into contributions from changes in invariant and specific feature components:

$$\frac{\partial f(x)}{\partial \theta_f} = \frac{\partial f_{\text{inv}}(x)}{\partial \theta_f} + \frac{\partial f_{\text{spec}}(x)}{\partial \theta_f} \tag{18}$$

This assumes that changes in $\theta_f$ can conceptually lead to changes in both domain-invariant and domain-specific aspects of the features $f(x)$.

**Step 2:** Adapter learns mainly from domain-specific signals. As motivated by IB (Eq. 12), the adapter $g$ primarily models domain-specific information. Its parameters $\theta_g$ are updated to capture domain-specific characteristics:

$$\frac{\partial \mathcal{L}}{\partial \theta_g} \approx \frac{\partial \mathcal{L}}{\partial F(x)} \cdot \frac{\partial g(f(x))}{\partial \theta_g} \tag{19}$$

**Step 3:** The encoder-decoder receives gradients influenced by both its direct path and the adapter path. Substituting Eq. 15 and Eq. 18 into Eq. 14:

$$\frac{\partial \mathcal{L}}{\partial \theta_f} = \frac{\partial \mathcal{L}}{\partial F(x)} \cdot \left[ \mathbf{I} + \frac{\partial g(f(x))}{\partial f(x)} \right] \cdot \left[ \frac{\partial f_{\text{inv}}(x)}{\partial \theta_f} + \frac{\partial f_{\text{spec}}(x)}{\partial \theta_f} \right] \tag{20}$$

**Step 4:** Adapter approximates the negative of the projected domain-specific representation. The adapter $g$ learns to model domain-specific aspects. We hypothesize $g(f(x)) \approx -f_{\text{spec}}(x)$. Using the projection, $g(f(x)) \approx -P_{\text{spec}} f(x)$. Thus, the Jacobian of the adapter function is (assuming $P_{\text{spec}}$ acts as an approximately linear operator or its dependency on $f(x)$ is negligible for this derivative):

$$\frac{\partial g(f(x))}{\partial f(x)} \approx -P_{\text{spec}} \tag{21}$$

**Step 5:** Substituting the adapter's Jacobian approximation:

$$\frac{\partial \mathcal{L}}{\partial \theta_f} \approx \frac{\partial \mathcal{L}}{\partial F(x)} \cdot [\mathbf{I} - P_{\text{spec}}] \cdot \left[ \frac{\partial f_{\text{inv}}(x)}{\partial \theta_f} + \frac{\partial f_{\text{spec}}(x)}{\partial \theta_f} \right] \tag{22}$$

**Step 6:** Expanding the terms involving the projection. Since $P_{\text{inv}} \approx \mathbf{I} - P_{\text{spec}}$, we have:

$$\frac{\partial \mathcal{L}}{\partial \theta_f} \approx \frac{\partial \mathcal{L}}{\partial F(x)} \cdot P_{\text{inv}} \cdot \left[ \frac{\partial f_{\text{inv}}(x)}{\partial \theta_f} + \frac{\partial f_{\text{spec}}(x)}{\partial \theta_f} \right] \tag{23}$$

Expanding this:

$$\frac{\partial \mathcal{L}}{\partial \theta_f} \approx \frac{\partial \mathcal{L}}{\partial F(x)} \cdot \left[ P_{\text{inv}} \frac{\partial f_{\text{inv}}(x)}{\partial \theta_f} + P_{\text{inv}} \frac{\partial f_{\text{spec}}(x)}{\partial \theta_f} \right] \tag{24}$$

**Step 7:** Assuming approximate orthogonality of gradient components. If changes to $f_{\text{inv}}(x)$ due to $\theta_f$ (i.e., $\frac{\partial f_{\text{inv}}(x)}{\partial \theta_f}$) lie in the invariant subspace, then:

$$P_{\text{inv}} \frac{\partial f_{\text{inv}}(x)}{\partial \theta_f} \approx \frac{\partial f_{\text{inv}}(x)}{\partial \theta_f} \tag{25}$$

If changes to $f_{\text{spec}}(x)$ due to $\theta_f$ (i.e., $\frac{\partial f_{\text{spec}}(x)}{\partial \theta_f}$) lie in the specific subspace (orthogonal to the invariant one), then:

$$P_{\text{inv}} \frac{\partial f_{\text{spec}}(x)}{\partial \theta_f} \approx 0 \tag{26}$$

Substituting Eq. 25 and Eq. 26 into Eq. 24:

$$\frac{\partial \mathcal{L}}{\partial \theta_f} \approx \frac{\partial \mathcal{L}}{\partial F(x)} \cdot \left[ \frac{\partial f_{\text{inv}}(x)}{\partial \theta_f} + 0 \right] = \frac{\partial \mathcal{L}}{\partial F(x)} \cdot \frac{\partial f_{\text{inv}}(x)}{\partial \theta_f} \tag{27}$$

This derivation shows that, under these assumptions, the gradient updates to the encoder-decoder parameters $\theta_f$ are predominantly guided by the domain-invariant components of the features. The adapter, by learning to approximate $-P_{\text{spec}} f(x)$, effectively filters or projects out the domain-specific gradient components that would otherwise update $\theta_f$. Consequently, the ED network is primarily optimized toward learning domain-invariant representations, while the adapter itself (via Eq. 16) focuses on domain-specific adaptation. This provides a theoretical foundation for viewing adapters as information decouplers.

## B. Centered Kernel Alignment (CKA)

CKA (Kornblith et al., 2019) is a widely used metric for comparing the similarity between two data representations by normalizing the Hilbert-Schmidt Independence Criterion (HSIC). This normalization mitigates the effects of scaling differences between the two representations.

## B.1. Dot Product-Based Similarity

CKA builds on the concept of dot product-based similarity, which relates inter-example similarities to feature similarities. Given two representations $\mathbf{X} \in \mathbb{R}^{n \times d}$ and $\mathbf{Y} \in \mathbb{R}^{n \times d}$, the dot product similarity between examples is:

$$\langle \text{vec}(\mathbf{X}\mathbf{X}^\top), \text{vec}(\mathbf{Y}\mathbf{Y}^\top) \rangle = \text{tr}(\mathbf{X}\mathbf{X}^\top \mathbf{Y}\mathbf{Y}^\top) = \|\mathbf{Y}^\top \mathbf{X}\|_F^2. \quad (28)$$

The terms $\mathbf{X}\mathbf{X}^\top$ and $\mathbf{Y}\mathbf{Y}^\top$ represent dot products between pairs of examples in $\mathbf{X}$ and $\mathbf{Y}$. The left-hand side measures similarity between these structures, while the right-hand side captures feature similarity through the squared dot products.

## B.2. Hilbert-Schmidt Independence Criterion (HSIC)

HSIC generalizes dot product-based similarity by measuring dependence between two sets of variables in reproducing kernel Hilbert spaces (RKHS). For centered matrices $\mathbf{X}$ and $\mathbf{Y}$, Equation (1) implies:

$$\frac{1}{(n-1)^2}\text{tr}(\mathbf{X}\mathbf{X}^\top \mathbf{Y}\mathbf{Y}^\top) = \|\text{cov}(\mathbf{X}^\top, \mathbf{Y}^\top)\|_F^2. \quad (29)$$

HSIC extends this to kernel methods, where $\mathbf{K}_{ij} = k(x_i, x_j)$ and $\mathbf{L}_{ij} = l(y_i, y_j)$ are kernel matrices. The empirical HSIC estimator is:

$$\text{HSIC}(\mathbf{K}, \mathbf{L}) = \frac{1}{(n-1)^2}\text{tr}(\mathbf{K}\mathbf{H}\mathbf{L}\mathbf{H}), \quad (30)$$

with $\mathbf{H}$ being the centering matrix:

$$\mathbf{H} = \mathbf{I}_n - \frac{1}{n}\mathbf{1}\mathbf{1}^\top, \quad (31)$$

where $\mathbf{I}_n$ is the identity matrix and $\mathbf{1}$ is a vector of ones. HSIC measures dependence between two distributions and converges to the population value at a rate of $1/\sqrt{n}$.

## B.3. Centered Kernel Alignment (CKA)

To address the scaling issues inherent in HSIC, CKA normalizes the dependence measure. CKA between two kernel matrices $\mathbf{K}$ and $\mathbf{L}$ is defined as:

$$\text{CKA}(\mathbf{K}, \mathbf{L}) = \frac{\text{HSIC}(\mathbf{K}, \mathbf{L})}{\sqrt{\text{HSIC}(\mathbf{K}, \mathbf{K}) \cdot \text{HSIC}(\mathbf{L}, \mathbf{L})}}. \quad (32)$$

The numerator measures the similarity between the two kernel matrices, while the denominator normalizes this similarity by accounting for the self-similarities within each representation. This normalization ensures that CKA is invariant to isotropic scaling, making it robust to variations in the magnitudes of the features.

We extract features from the source and target domains through the feature extractor, treat them as two sets of representations, and compute the domain similarity using the CKA formula above.

## C. Adapter Naturally Serves as Decoupler

The deeper the neural network is, the more semantic information its features can encompass, meaning it can better represent category information. Different categories exhibit more distinctive semantic information at higher layers. We validate it for CKA measure in paper, here, we further illustrate this perspective by visualizing feature maps. As shown in Fig.11, the two examples illustrate that as the depth of the layers increases, the model focuses more on specific objects, extracting features that are more discriminative.

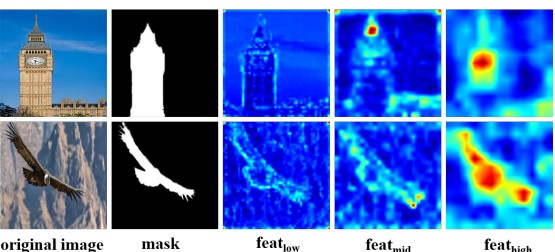

original image    mask    feat$_{low}$    feat$_{mid}$    feat$_{high}$

Figure 11: The deeper the neural network is, the more semantic information its features can encompass.

The adapter is a network layer positioned deeper than each layer of the backbone network, placing itself at a relatively deeper level while the backbone is relatively shallower. Intuitively, the adapter serves as a natural boundary line for domain features and always captures a level of semantics higher than what the backbone learns, making it more specific. We validate it for CKA measure in paper, here, we further illustrate this perspective by visualizing feature maps at different levels, and the results are shown in Figure.12. For each column, after attaching an adapter, these feature maps contain more specific features like the bird's profile compared with the first row, which means the adapter assimilates domain-specific information.

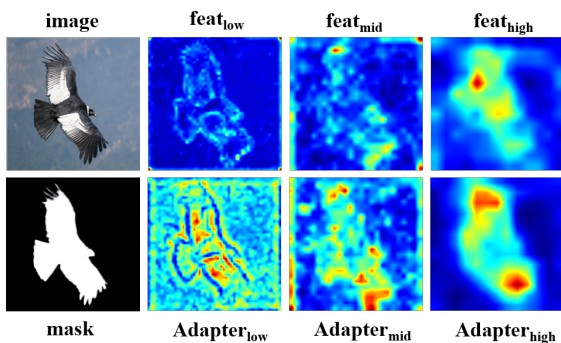

image    feat$_{low}$    feat$_{mid}$    feat$_{high}$

mask    Adapter$_{low}$    Adapter$_{mid}$    Adapter$_{high}$

Figure 12: The adapter serves as a natural boundary line for domain features and always captures a level of semantics higher than what the backbone learns

## D. SAM: Flatten Loss Landscape

For any $\rho > 0$ and any distribution $\mathscr{D}$, with probability $1 - \delta$ over the choice of the training set $\mathcal{S} \sim \mathscr{D}$,

$$L_{\mathscr{D}}(\boldsymbol{w}) \leq \max_{\|\boldsymbol{\epsilon}\|_2 \leq \rho} L_{\mathcal{S}}(\boldsymbol{w} + \boldsymbol{\epsilon}) + \tag{33}$$

$$\sqrt{\frac{k \log\left(1 + \frac{\|\boldsymbol{w}\|_2^2}{\rho^2}\left(1 + \sqrt{\frac{\log(n)}{k}}\right)^2\right) + \frac{4 \log \frac{n}{\delta} + \tilde{O}(1)}{n-1}}{n-1}} \tag{34}$$

where $n = |\mathcal{S}|$, $k$ is the number of parameters and we assumed $L_{\mathscr{D}}(\boldsymbol{w}) \leq \mathbb{E}_{\epsilon_i \sim \mathcal{N}(0,\rho)}[L_{\mathscr{D}}(\boldsymbol{w} + \boldsymbol{\epsilon})]$. This condition implies that adding Gaussian perturbations should not reduce the test error, which generally holds for the final solution but not necessarily for all $\boldsymbol{w}$.

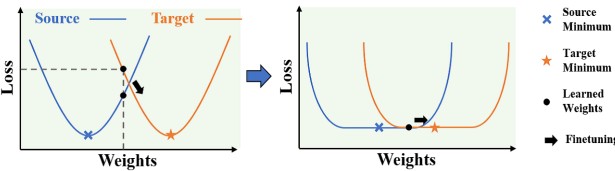

Figure 13: Sharpness-Aware Minimization (SAM): flattens the loss landscape of the DFN, limits its absorption of excessive source-domain information (prevent overfitting to the source-domain samples), thereby facilitating the fine-tuning of DFN during target-domain stage.

The overview of the main text, we present the process diagram of SAM and its effects. Here, we provide further detailed explanations, as shown in Figure. 13. During source domain training, the DFN absorbs domain-specific information, leading to loss minima that are specific to the source domain, i.e., source minima. When domain shift occurs, the weights learned by the DFN become misaligned on the target domain. If the loss landscape of the DFN is too sharp, fine-tuning to the target minima becomes more difficult. Adopting SAM during training on the source domain flattens the loss landscape of the DFN, making it easier to fine-tune to the target minima when transferring to the target domain.

## E. The Analysis of Hyper-parameter

Fine-tuning of DFN is performed using the Adam optimizer, with learning rates set at 1e-3 for FSS-1000, 5e-1 for Deepglobe, 5e-3 for ISIC, and Chest X-ray. Each task undergoes a total of 50 iterations. As shown in Fig.14, we evaluated the fine-tuning performance on four target datasets at different learning rates. We also present experiments on the hyperparameter $\rho$ for Sharpness-Aware Minimization (SAM) as shown in table 12 (left).

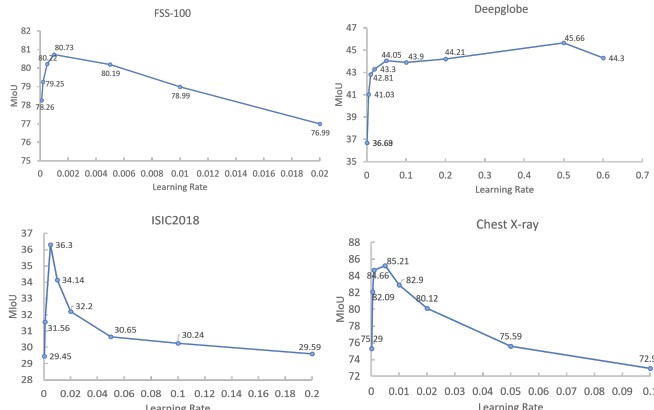

Figure 14: The Learning rate for the fine-tuning of four target datasets.

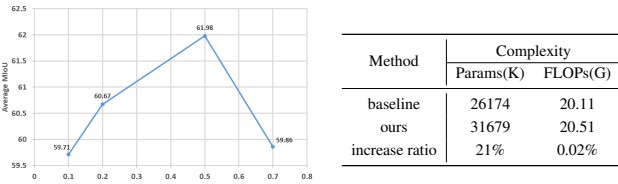

Table 12: **(Left)** The hyper-parameter $\rho$ of the SAM. **(Right)** The complexity analysis.

| Method | Complexity | |
|---|---|---|
| | Params(K) | FLOPs(G) |
| baseline | 26174 | 20.11 |
| ours | 31679 | 20.51 |
| increase ratio | 21% | 0.02% |

## F. The Analysis of Computational Complexity

As shown in Table 12 (right), we present the results of the complexity analysis, demonstrating that our method incurs minimal overhead in terms of parameter count and training time. We use a single 4090 GPU for training and testing.

**The Computational Efficiency of SAM-SVN**  Our SAM-SVN is used only during source domain training and not during fine-tuning or inference, so it does not affect the computational efficiency during inference. Regarding efficiency during training in the source domain, although it requires double backpropagation, it is applied only to the singular value matrix of DFN, resulting in negligible additional computation. As shown in Table 13, we demonstrate the computational efficiency of SAM-SVN by measuring the efficiency of the baseline, PATNet (which adopts the same baseline as ours), SAM applied to the entire model, SAM applied to DFN, and SAM applied to SVN (the singular value matrix of DFN).

| Method | Baseline | PATNet | SAM-Whole | SAM-DFN | SAM-SVN |
|---|---|---|---|---|---|
| **FLOPs (G)** | 20.11 | 22.62 | 26.82 | 22.68 | 20.51 |
| **Increase Ratio** | – | 12.4% | 33.35% | 12.77% | **1.99%** |

Table 13: Computational efficiency comparison of different SAM variants.

## G. More Ablation Study Results

In the main text, we presented the average mIoU across four target datasets to demonstrate our design effectiveness. Table 14 provides detailed performance results for each individual target dataset. While the main ablation studies used ResNet-50 as the backbone, we supplement these findings with additional experiments using ViT-Base (following FPTrans (Zhang et al., 2022), Table 15).

| DFN | SAM | SVD | FSS-1000 | | Deepglobe | | ISIC | | Chest X-ray | |
|-----|-----|-----|--------|--------|--------|--------|--------|--------|--------|--------|
| | | | 1-shot | 5-shot | 1-shot | 5-shot | 1-shot | 5-shot | 1-shot | 5-shot |
| | | | 77.53 | 80.99 | 29.65 | 35.08 | 31.20 | 35.10 | 51.88 | 54.36 |
| ✓ | | | 80.67 | 84.87 | 41.29 | 44.37 | 34.82 | 48.04 | 82.78 | 89.07 |
| ✓ | ✓ | | **80.97** | 85.25 | 42.28 | 45.52 | 35.21 | 50.12 | 84.14 | 90.08 |
| ✓ | ✓ | ✓ | 80.73 | **85.80** | **45.66** | **47.98** | **36.30** | **51.13** | **85.21** | **90.34** |

Table 14: (**ResNet50**): Ablation study on various designs, observing the MIoU for 1-shot and 5-shot on four datasets.

| DFN | SAM | SVD | FSS-1000 | | Deepglobe | | ISIC | | Chest X-ray | |
|-----|-----|-----|--------|--------|--------|--------|--------|--------|--------|--------|
| | | | 1-shot | 5-shot | 1-shot | 5-shot | 1-shot | 5-shot | 1-shot | 5-shot |
| | | | 78.90 | 81.77 | 38.29 | 42.54 | 47.60 | 52.90 | 78.92 | 80.61 |
| ✓ | | | 81.83 | 83.86 | 39.12 | 46.23 | 48.96 | 56.79 | 81.97 | 85.32 |
| ✓ | ✓ | | 82.76 | 84.59 | 39.37 | 46.98 | 50.09 | 57.51 | 82.77 | 86.15 |
| ✓ | ✓ | ✓ | **82.97** | **85.72** | **39.45** | **47.67** | **50.36** | **58.53** | **83.18** | **87.14** |

Table 15: (**ViT-Base**): Ablation study on various designs, observing the MIoU for 1-shot and 5-shot settings.

We futher investigate how our method impacts the encoder's output. Using the relative CKA similarity (the ratio of CKA between Pascal and target datasets to Pascal itself, i.e., $\text{CKA}/\text{CKA}_{pascal}$). A higher value indicates a lower domain similarity, meaning the encoder produces more agnostic features. Figure 15 reveals that attaching DFN to the backbone increases relative CKA, and further improvement occurs after applying SAM-SVN. This suggests the model has acquired more agnostic representations, reducing the distance between domains.

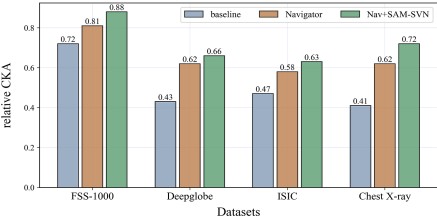

Figure 15: Our designs increase encoder's domain similarity, enhancing the model's learning of agnostic knowledge.

## H. Comparison with More Methods under Their Settings

Because the baseline SSP (Fan et al., 2022) used by the IFA (Nie et al., 2024) aggregates all samples within a batch when computing the foreground and background prototypes,

existing CD-FSS methods set the batch size to 1 during testing. This is to avoid unfair comparisons that arise from including information from other samples in the batch. However, IFA sets the batch size to 96 during testing. To ensure a fair comparison, we conduct comparisons with IFA using a batch size of 96.

| Method | FSS-1000 | | Deepglobe | | ISIC | | Chest X-ray | |
|--------|--------|--------|--------|--------|--------|--------|--------|--------|
| | 1-shot | 5-shot | 1-shot | 5-shot | 1-shot | 5-shot | 1-shot | 5-shot |
| IFA | 80.1 | 82.4 | 50.6 | 58.8 | 66.3 | 69.8 | 74.0 | 74.6 |
| **Ours** | 82.6 | 87.9 | 51.3 | 59.2 | 68.5 | 71.4 | 86.1 | 91.6 |

Table 16: Comparison with IFA under it specific testing setting. (Backbone: Res-50, Batch size: 96)

## I. Related Work

**Few-shot learning** Few-shot learning focuses on developing robust representations for novel concepts with limited annotated samples (An et al., 2024a;b). Existing approaches can be broadly categorized into three primary frameworks: metric learning methods (Snell et al., 2017; Vinyals et al., 2016), optimization-based methods (Finn et al., 2017), and graph-based methods (Garcia & Bruna, 2017). Cross-domain few-shot learning (Guo et al., 2020; Zou et al., 2024b) is receiving increasing interest recently, there exist disparities not only in the data distribution but also in the label space between the meta-testing stage and the meta-training stage. A significant challenge in this field remains effectively bridging the domain gap between source and target domains when only a few labeled samples are available.

## J. More Dataset Details

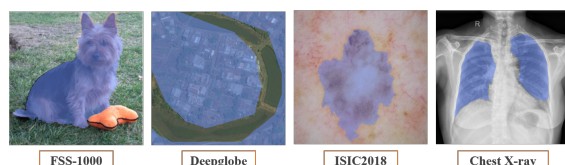

Figure 16: Example of segmentation for four datasets.

We adopt the benchmark (see Figure.16) proposed by PAT-Net (Lei et al., 2022) and follow the same data preprocessing procedures as the dataset it employs.

**PASCAL**-$5^i$ (Shaban et al., 2017) is an extended version of PASCAL VOC 2012 (Everingham et al., 2010), incorporating supplementary annotation enhancement details from the SDS dataset (Hariharan et al., 2011). We employ PASCAL-$5^i$ as our source domain for training. Subsequently, we assess the performance of the trained models across four additional datasets.

**FSS-1000** (Li et al., 2020) is a natural image dataset for few-shot segmentation, containing 1000 categories, and each

category has 10 samples. We follow the official split for semantic segmentation in our experiment and present results on the designated testing set, consisting of 240 classes and 2,400 images. We regard FSS-1000 as a target domain for testing.

**Deepglobe** (Demir et al., 2018) comprises satellite images with dense pixel-level annotations across 7 categories: urban, agriculture, rangeland, forest, water, barren, and unknown. As ground-truth labels are only provided in the training set, we rely on the official training dataset, consisting of 803 images, to present our results. We adopt it as our target domain for testing and we take the same processing approach as PATNet.

**ISIC2018** (Codella et al., 2019; Tschandl et al., 2018) is designed for skin cancer screening, and comprises lesion images, with each image containing precisely one primary lesion. The dataset is processed and utilized in accordance with the standards set by PATNet. And we regard ISIC2018 as a target domain for testing.

**Chest X-ray** (Candemir et al., 2013; Jaeger et al., 2013) is an X-ray dataset for Tuberculosis, consisting of 566 images ($4020 \times 4892$ resolution). These images are derived from 58 Tuberculosis cases and 80 normal cases. To address large image sizes, a common practice involves downsizing to $1024 \times 1024$ pixels.

