# OpenReview forum: "Adapter Naturally Serves as Decoupler for Cross-Domain Few-Shot Semantic Segmentation"
_ICML.cc/2025/Conference — ICML 2025 spotlightposter_

### Official Review · Reviewer_eLxN · 2025-03-11

**Overall Recommendation:** 4

**Summary:**

The paper proposes utilizing an adapter for cross-domain few-shot semantic segmentation. The authors first demonstrate that the adapter naturally serves as a decoupler, and then design a DFN network to decouple source domain information into domain-agnostic and domain-specific components. They also propose SAM-SVN to mitigate potential overfitting issues of DFN on the source samples, achieving good performance.

**Claims And Evidence:**

The claims made in the paper is that the adapter naturally serves as a decoupler for domain-specific information. The authors provide detailed experiments and analysis to support this claim.

**Essential References Not Discussed:**

The following papers are related to the model’s hypercorrelation design and should be cited:

- CVPR 2024, Rethinking Few-shot 3D Point Cloud Semantic Segmentation
- ICLR 2025, Multimodality Helps Few-shot 3D Point Cloud Semantic Segmentation

**Experimental Designs Or Analyses:**

The main experiments are comprehensive and demonstrate the effectiveness of the proposed method. However, since SAM-SVN requires an additional forward-backward computation that may be computationally expensive, it would be useful to compare the model's computational efficiency with other baselines.

**Methods And Evaluation Criteria:**

The proposed method is well explained, and the evaluation criteria and datasets used are appropriate for the task.

**Other Comments Or Suggestions:**

Please see the Questions.

**Other Strengths And Weaknesses:**

The paper is clearly written and easy to follow. The motivations for the design choices are clear and reasonable, and the experiments demonstrate the superior performance achieved by the proposed method.

**Questions For Authors:**

Since SAM-SVN requires an additional forward-backward computation that may be computationally expensive, could you compare the model's parameter count and computational efficiency with other baselines?

**Relation To Broader Scientific Literature:**

The proposed designs complement the broader literature and introduce new designs.

**Theoretical Claims:**

There are no theoretical claims.

---

> ### Author Rebuttal · Authors · 2025-04-01
>
> ## 1. The computational efficiency of SAM-SVN：
>
> Our SAM-SVN is used only during source domain training and not during fine-tuning or inference, so it does not affect the computational efficiency during inference. Regarding efficiency during training in the source domain, although it requires double backpropagation, it is applied only to the singular value matrix of DFN, resulting in negligible additional computation. We demonstrate the computational efficiency of SAM-SVN by measuring the efficiency of the baseline, PATNet (which adopts the same baseline as ours), SAM applied to the entire model, SAM applied to DFN, and SAM applied to SVN (singular value matrix of DFN). The results are shown in the table below:
>
> |                | baseline | PATNet [1] | SAM-Whole | SAM-DFN |  SAM-SVN  |
> | -------------- | :------: | :--------: | :-------: | :-----: | :-------: |
> | FLOPs(G)       |  20.12   |   22.63    |   26.83   |  22.69  |   20.52   |
> | increase ratio |    /     |   12.4%    |  33.35%   | 12.77%  | **1.99%** |
>
> [1] Cross-Domain Few-Shot Semantic Segmentation
>
> ## 2. References：
>
> We promise to cite the listed references in the final version.

---

> > ### Comment · Reviewer_eLxN · 2025-04-03
> >
> > Thank you for the response. My concerns have been addressed and I would update my recommendation to accept.

---

> > > ### Author Response · Authors · 2025-04-04
> > >
> > > We sincerely appreciate your response and recognition of our rebuttal.

---

### Official Review · Reviewer_iPWG · 2025-03-11

**Overall Recommendation:** 4

**Summary:**

This paper find an interesting phenomenon that a sort of adapters naturally serve as domain-information decoupler for the CDFSS task. By comprehensive experiments, the authors validate the condition that makes adapters to be decouplers. Then, they extend such a natural decoupler by sharpness-aware minimization to build the DFN for CDFSS, which shows state-of-the-art performance.

**Claims And Evidence:**

The phenomenon that adapters naturally serve as decoupler is novel to me. Most claims are validated in the experiments. I like the writing of this paper to detailly study each component of the phenomenon and methods.

However, this paper claims the decoupler phenomenon for CDFSS methods, but it majorly studies the HSNet structure. Nowadays many other structures are available to achieve state-of-the-art performance, such as [1] and [2]. Indeed, HSNet is prevailing, but I wish to see whether this phenomenon and method could fit other structures.

[1] Lightweight Frequency Masker for Cross-Domain Few-Shot Semantic Segmentation
[2] Feature-Proxy Transformer for Few-Shot Segmentation

**Essential References Not Discussed:**

I think the authors can supplement the related work with some latest work about adapters such as [1], although some of the most famous ones have been studied in section 2.

[1] Lightweight Frequency Masker for Cross-Domain Few-Shot Semantic Segmentation

**Experimental Designs Or Analyses:**

The experiments and analysis are comprehensive and thorough. I appreciate the author to validate each aspect of this problem and design choice.

However, I wonder how the proposed method influence the source-domain performance. Since the adapter decouples domain information which is important for the source domain, will it harm the source-domain performance? I know the FSS dataset is close to Pascal, but I wish to see the experiments on Pascal.

**Methods And Evaluation Criteria:**

The proposed method is a kind of adapter found to decouple source-domain information, and a training strategy to resist the overfitting, which are reasonable and novel to me. The performance in table 6 is good compared with current works.
However, in table 6, I see resnet50 and vit-base achieves state-of-the-art performance differently on these four datasets. Similarly, in appendix, table 12 and 13 show that the proposed method contribute differently to resnet and vit. Could you give some insights on these results?

**Other Comments Or Suggestions:**

Typos: Section D title, whit -> with.

**Other Strengths And Weaknesses:**

I see in section 2 the authors use CKA to validate the domain similarity, but in section 4 the MMD is used for validation. Although the CKA experiments are included in the appendix, I still suggest keeping the criteria consistent in the paper.

**Questions For Authors:**

I wonder what will happen if a trained adapter is removed from network. That is, after the source-domain training, since adapters grab domain information that is harmful for target domains, how is the performance if we directly remove these adapters and only keep the remaining structures and weights for segmentation? Will it be higher than appending the source-domain-trained adapter but not finetuning it? Furthermore, during the target-domain finetuning, how is the performance if we directly use a scratch adapter for finetuning? These experiments can help understand the information captured by the source-domain-trained adapter.

**Relation To Broader Scientific Literature:**

This work can be applied to medical analysis (e.g., ISIC for skin diseases, chest x-ray for lung diseases), so it will help the scientific study by providing a easy-to-adapt model.

**Theoretical Claims:**

The theoretical side of this paper is in the sharpness-aware minimization. Experiments validate the influence that DFN brings to the sharpness.
However, do other choice of adapters also influence the sharpness? I suggest adding these experiments to validate from the sharpness side that only the DFN can decouple domain information.

---

> ### Author Rebuttal · Authors · 2025-04-01
>
> ## 1. Our method can fit different structures：
> Our structure and the APM[1] structure are both based on HSNet, while our ViT structure is based on FPTrans[2]. Additionally, in the appendix, we used the SSP-based architecture for comparison with IFA. Thus, our method can be applied to networks with various architectures.
>
> ## 2. The reason for the different performance between ResNet and ViT：
> The difference is due to the datasets' bias towards local recognition cues or global ones.
>
> We use KL divergence (DisFB) to measure the similarity between the foreground and background of datasets. A higher DisFB suggests a greater requirement for the model's global resolution capability, with ISIC having the highest DisFB. ResNet, due to its convolution-based design, has local prior characteristics, making it more effective for datasets like DeepGlobe that demand detailed local features. On the other hand, ViT, based on attention mechanism, excels in long-range dependencies and global resolution, offering better performance for datasets like ISIC that rely more on global perception.
> |       | FSS1000 | Deepglobe | ISIC  | ChestX |
> | ----- | :-----: | :-------: | :---: | :----: |
> | DisFB |  0.113  |   0.125   | 0.156 | 0.131  |
>
> ## 3. Validating adapter choice impact on sharpness
>
> In the main text, we measured loss fluctuations by adding Gaussian noise to observe the changes in the sharpness of loss landscapes before and after integrating DFN. Here, we further measure the impact of different adapter choices on sharpness. Consistent with Fig3 and Fig5 in the main text, we analyze based on position and structure (res: residual, ser: serial, BKB: backbone, enc-dec: encoder-decoder):
>
> |  Position  | Baseline | BKB shallower | BKB deeper | Between enc-dec |
> | :--------: | :------: | :-----------: | :--------: | :-------------: |
> | loss fluc. |  0.398   |     0.402     |   0.521    |      0.405      |
>
> | Structure  | Baseline | conventional+res | LoRA+res | conventional+ser |
> | :--------: | :------: | :--------------: | :------: | :--------------: |
> | loss fluc. |  0.398   |      0.521       |  0.533   |      0.399       |
>
> The results above show that, from the perspective of sharpness, significant changes in loss fluctuations occur only when residual links are satisfied and the position is deep within the backbone (with the adapter structure not being a determining factor). This indicates that the adapter has captured domain-specific related information, which is consistent with the conclusions drawn in the main text.
>
> ## 4. Our methods can benefit general few-shot segmentation tasks
>
> We tested the performance of DFN on the source domain (Pascal). Pascal consists of 20 classes and is set to a 4-fold configuration in the FSS setup. This means training is conducted on 5 classes, while testing is performed on 15 classes that were not seen during the training phase. Due to its ability to enhance the model's adaptation to new domains/classes, it also provides benefits for general few-shot segmentation. After fine-tuning DFN, its positive impact becomes even more pronounced.
>
> | 1shot（Pascal） | Fold0 | Fold1 | Fold2 | Fold3 | Mean |
> | --------------- | :---: | :---: | :---: | :---: | :--: |
> | Baseline        | 64.3  | 70.7  | 60.3  | 60.5  | 64.0 |
> | DFN w/o ft      | 65.2  | 70.9  | 60.8  | 61.3  | 64.6 |
> | DFN w/ ft       | 66.8  | 72.4  | 62.5  | 62.7  | 66.1 |
>
> ## 5. More settings to DFN：
> Here, we include more experiments on the DFN training and fine-tuning settings：1）DFN is involved in source domain training but removed in the target domain. 2）DFN is involved in source domain training, but its original parameters are discarded and it learns from scratch in the target domain (using different initialization methods).
> | 1-shot | FSS1000 | Deepglobe | ISIC  | ChestX | Mean |
> | :------: | :-----: | :-------: | :---: | :----: | ----- |
> |baseline|77.53|29.65|31.20|51.88|47.57|
> |DFN (remove in target)|78.16|38.21|34.12|76.92|56.85|
> |DFN(scratch, kaiming init)|79.02|42.53| 36.63 | 82.03  |60.05|
> |DFN(scratch, xavier init)|79.83|45.57| 34.79|79.46|59.91|
> |DFN(random gauss init)|78.97|38.75| 33.82 |78.59|57.53|
> |DFN|80.73|45.66| 36.30|85.21|61.98|
>
>
> For the setting that DFN is removed in the target domain, DFN captures domain-specific information during source domain training and guides the model to learn domain-agnostic knowledge, resulting in a significant performance improvement compared to the baseline. However, due to a lack of target-specific knowledge, this approach is suboptimal. In the setting that DFN learns from scratch, its performance is influenced by the initialization method, yet it still performs well under various initializations. Our approach (i.e., DFN+SAM-SVN) is seen as using DFN to guide the model to focus on domain-agnostic knowledge in the source domain, while simultaneously providing DFN with a reasonable initialization value beneficial for adapting to various domains. Therefore, it achieves the best performance.

---

### Official Review · Reviewer_U2qX · 2025-03-12

**Overall Recommendation:** 3

**Summary:**

The paper proposes that adapters naturally serve as domain information decouplers in Cross-Domain Few-Shot Segmentation (CD-FSS) by separating domain-specific and domain-agnostic features. Based on this insight, the authors introduce Domain Feature Navigator (DFN), a structure-based decoupler that captures domain-specific knowledge without requiring explicit domain losses. To prevent DFN from overfitting to source samples, they propose SAM-SVN, which applies Sharpness-Aware Minimization (SAM) on Singular Value Norms (SVN) to constrain overfitting while preserving domain decoupling. The approach is evaluated on FSS-1000, DeepGlobe, ISIC, and Chest X-ray datasets, showing gains over SOTA methods. Ablation studies confirm that DFN improves domain decoupling, while SAM-SVN enhances generalization by reducing loss sharpness. Qualitative results demonstrate that DFN redirects model attention toward domain-invariant features, improving segmentation.

**Claims And Evidence:**

The paper provides strong empirical evidence supporting its claims through quantitative results, ablation studies, and qualitative visualizations. The effectiveness of DFN as a domain decoupler is demonstrated through domain similarity CKA analysis.

However, some potential weaknesses exist:
(1) The claim that DFN naturally serves as a domain decoupler is supported by empirical observations (CKA similarity changes) but lacks a theoretical foundation explaining why this occurs.

(2) While SAM-SVN is shown to reduce sharpness and improve generalization, the trade-off between decoupling and domain knowledge retention is not deeply analyzed.

**Essential References Not Discussed:**

No, just one related work that also found residual connect is important, aligns with the finding of this paper.

Wang, Pei, Yijun Li, and Nuno Vasconcelos. "Rethinking and improving the robustness of image style transfer." Proceedings of the IEEE/CVF conference on computer vision and pattern recognition. 2021.

**Experimental Designs Or Analyses:**

Yes, I reviewed the experimental design and analyses, which are generally well-structured and sound.

**Methods And Evaluation Criteria:**

Yes, both the proposed methods and evaluation criteria are well-aligned in my opinion.

**Other Comments Or Suggestions:**

Figure 1 is hard to understand. So half of feature maps is to learn domain-agnostic knowledge and half for domain-specific knowledge? Exactly half? Any references? What is DFN? Maybe need to clarify in the caption.

Line 95, I can feel what the authors want to deliver, but hope to see more explanations why the decrease means the adapter captures domain-specific information


Table 1, is the relative or absolute change can hint something like for FSS-1000, only 0.0024 increase but for Deepglobe, it is almost 0.02.

The analysis relies on CKA, is it trustworthy? Any other metric to use for reference?

**Other Strengths And Weaknesses:**

No

**Questions For Authors:**

No

**Relation To Broader Scientific Literature:**

The paper builds on prior work in Cross-Domain Few-Shot Segmentation (CD-FSS), domain adaptation.

**Theoretical Claims:**

The paper does not present formal theoretical proofs but instead supports its claims through empirical observations and experimental validation.

---

> ### Author Rebuttal · Authors · 2025-04-01
>
> ## 1. Deeper theoretical analysis for “natural decoupling”：
>
> Due to space limitation, please refer to reviewer ueDb's reply 1 for theoretical analysis.
>
>
> ## 2. Analysis of trade-off between decoupling and domain knowledge retention:
>
> For any $\rho>0$ and any distribution $\mathscr{D}$, with probability $1-\delta$ over the choice of the training set $\mathcal{S}\sim \mathscr{D}$, SAM bounds the generalization error as
> $$
> L\_\mathscr{D}(\boldsymbol{w}) \leq \max\_{\|\boldsymbol{\epsilon}\|\_2 \leq \rho} L\_\mathcal{S}(\boldsymbol{w} + \boldsymbol{\epsilon}) + \sqrt{\frac{k\log\left(1+\frac{\|\boldsymbol{w}\|\_2^2}{\rho^2}\left(1+\sqrt{\frac{\log(n)}{k}}\right)^2\right)+ \frac{4\log\frac{n}{\delta} + \tilde{O}(1)}{n-1}}{n-1}}
> $$
> where $n=|\mathcal{S}|$, $k$ is the number of parameters, and we assume $L\_\mathscr{D}(\boldsymbol{w}) \leq \mathbb{E}\_{\epsilon\_i \sim \mathcal{N}(0,\rho)}[L\_\mathscr{D}(\boldsymbol{w}+\boldsymbol{\epsilon})]$. The trade-off is controlled by $\rho$. For $\rho$, a larger value indicates more perturbations are added, emphasizing domain knowledge retention (experiment is in Appendix Tab11 left)；For $w$​, a larger perturbation range signifies a greater focus on knowledge retention (experiment is in Tab7 right). All experiments regarding SAM-SVN are presented in Tab7 left, Fig10, Tab8, and Tab11 left.
>
>
> ## 3. Clarity for Figure 1:
>
> In fact, the feature does not contain domain-agnostic and domain-specific knowledge in equal parts; our diagram merely indicates that the feature includes both specific and agnostic information. Regarding DFN, which is a structure-based decoupler rather than a loss-based one like current approaches, it captures domain-specific information, thereby directing the model’s attention towards domain-agnostic knowledge (described in abstract and discussed in section 2). We will add a brief explanation of DFN in the caption to make it clearer for readers. Thank you very much for your suggestion.
>
>
>
> ## 4. More explanations for why cka decrease means "domain-specific"
>
> CKA is a metric used to measure domain similarity by comparing the distances between kernel centers of two sets of data. If the kernel centers of the representations extracted by neural networks from the two data sets are closer (resulting in a higher CKA value), it indicates that their data distributions in the feature space are more consistent, meaning their feature representations are more similar and they share more patterns. Conversely, a lower CKA value implies lower domain similarity, with the kernel centers being farther apart in the feature space. Lower domain similarity suggests that the extracted features are less similar, indicating that the two sets of representations share fewer common patterns and have more characteristics specific to their own data distributions (more domain-specific).
>
> To verify it is the domain gap that influences CKA, we divided the data in the Pascal into 20 groups based on categories and measured the CKA between different groups. We reported the CKA between the two most divergent groups, as well as the mean CKA and standard deviation (std) CKA. The results are as follows: It can be seen that even for the most divergent feature groups within the same dataset, the CKA is above 0.85, indicating the validity of using CKA to measure domain similarity.
>
> |  CKA   | max diff |  mean  |  std   |
> | :----: | :------: | :----: | :----: |
> | Pascal |  0.8656  | 0.8895 | 0.0179 |
>
>
>
> ## 5. Reliability of the CKA metric
>
> Absolute change is used here. We choose CKA as a measurement metric for several reasons: 1) CKA employs kernel center alignment, which eliminates measurement errors caused by extreme feature representations (noise); 2) Compared to other metrics like MMD and cosine similarity, CKA removes the effects of scaling differences; 3) CKA is more sensitive to differences in data distribution, allowing it to more accurately reflect distribution changes (detailed theory in the appendix).
>
> The discrepancy in change between FSS and DeepGlobe is consistent due to: 1) CKA's sensitivity to data distribution differences; 2) The smaller domain gap between FSS and Pascal (both being natural image datasets) results in smaller changes, whereas the larger domain gap between Deepglobe (remote sensing images) and Pascal leads to greater gains and changes.
>
> Additional metrics: In the experimental section (Figure 8), we also used MMD to validate our viewpoint, which is consistent with CKA. Furthermore, in the appendix, we employed relative CKA to mitigate the impact of data distribution (Figure 15).

---

### Official Review · Reviewer_ueDb · 2025-03-18

**Overall Recommendation:** 3

**Summary:**

This paper introduces a novel perspective on using adapters as structural domain decouplers for cross-domain few-shot semantic segmentation (CD-FSS). They introduce the Domain Feature Navigator (DFN), a structure-based decoupler inserted into deeper network layers with residual connections, and SAM-SVN, a sharpness-aware regularization method applied to the singular values of DFN weights to prevent overfitting.

**Claims And Evidence:**

good

**Essential References Not Discussed:**

A recent work on cd-fss is not discussed: SAM-Aware Graph Prompt Reasoning Network for Cross-Domain Few-Shot Segmentation, aaai 2025

**Experimental Designs Or Analyses:**

good

**Methods And Evaluation Criteria:**

good

**Other Comments Or Suggestions:**

please see weakness

**Other Strengths And Weaknesses:**

Strengths:
1. The discovery that adapters structurally decouple domain information (without explicit loss functions) is innovative and challenges existing loss-based domain adaptation paradigms.
2. SAM-SVN effectively balances domain-specific knowledge absorption and overfitting prevention.
3. Experiments demonstrate state-of-the-art performance on four benchmarks. Comprehensive ablation studies validate design choices (adapter position, residual connections, SAM-SVN). Experiments span multiple datasets (FSS-1000, DeepGlobe, ISIC, Chest X-ray) and backbones (ResNet-50, ViT), demonstrating robustness.

Weaknesses
1. The term "natural decoupling" lacks intuitive explanation. While experiments show reduced CKA similarity, a deeper theoretical analysis (e.g., information bottleneck principles) could strengthen claims. The rationale for perturbing singular values (vs. other parameters) in SAM-SVN needs further justification.
2. Comparisons with IFA use different batch sizes (96 vs. 1). Why?
3. Why is there an impact statement analysis on page 9? Does it exceed the page limit?

**Questions For Authors:**

please see weakness

**Relation To Broader Scientific Literature:**

good

**Theoretical Claims:**

good

---

> ### Author Rebuttal · Authors · 2025-04-01
>
> ## 1. Deeper theoretical analysis for “natural decoupling”：
>
> The behavior of adapters as decouplers can be analyzed through the Information Bottleneck (IB) theory. The IB objective is:
>
> $\mathcal{L}_{IB} = I(X;Z) - \beta I(Z;Y)$
>
> where $I(\cdot;\cdot)$ is mutual information, X is the input, Y is the final output label, $Z$ is the intermediate representation, and $\beta$ is the balance parameter. Decomposing the input as $X = X_{inv} + X_{spec}$, the IB objective becomes:
>
> $\mathcal{L}_{IB} = I(X\_{inv} + X\_{spec};Z) - \beta I(Z;Y)$
>
> For encoder-decoder network (ED) parameters $\theta_f$ and adapter parameters $\theta_g$ where $\theta_f \gg \theta_g$, the adapter's information capacity is much smaller than the ED, causing it to selectively absorb source domain-specific information that better optimizes the current training objective, resulting in $I(X_{spec};Z) \gg I(X_{inv};Z)$, which promotes gradient flow differentiation. In a network $f$ with residual adapter $g$, the forward propagation is: $F(x) = f(x) + g(f(x))$
>
> For ED parameters $\theta_f$, the gradient is: $\frac{\partial \mathcal{L}}{\partial \theta_f} = \frac{\partial \mathcal{L}}{\partial F(x)} \cdot \frac{\partial F(x)}{\partial f(x)} \cdot \frac{\partial f(x)}{\partial \theta_f}$, expanding the middle term: $\frac{\partial F(x)}{\partial f(x)} = I + \frac{\partial g(f(x))}{\partial f(x)}$
>
> where $I$ is the identity matrix (from the direct residual path), and the second term is the Jacobian matrix of the adapter function.
>
> For adapter parameters $\theta_g$, the gradient is proved to be $\frac{\partial \mathcal{L}}{\partial \theta_g} = \frac{\partial \mathcal{L}}{\partial F(x)} \cdot \frac{\partial g(f(x))}{\partial \theta_g}$
>
> Differentiated gradients: Due to the adapter's selective absorption of domain-specific information and the residual adapter structure, gradient flow naturally separates network optimization into two complementary learning objectives.
>
>
>
> ## 2. Further justification for perturbing singular values in SAM-SVN:
>
> For any $\rho>0$ and any distribution $\mathscr{D}$, with probability $1-\delta$ over the choice of the training set $\mathcal{S}\sim \mathscr{D}$, SAM bounds the generalization error as
> $$
> L\_\mathscr{D}(\boldsymbol{w}) \leq \max\_{\|\boldsymbol{\epsilon}\|\_2 \leq \rho} L\_\mathcal{S}(\boldsymbol{w} + \boldsymbol{\epsilon}) + \sqrt{\frac{k\log\left(1+\frac{\|\boldsymbol{w}\|\_2^2}{\rho^2}\left(1+\sqrt{\frac{\log(n)}{k}}\right)^2\right)+ \frac{4\log\frac{n}{\delta} + \tilde{O}(1)}{n-1}}{n-1}}
> $$
> where $n=|\mathcal{S}|$, $k$ is the number of parameters, and we assume $L\_\mathscr{D}(\boldsymbol{w}) \leq \mathbb{E}\_{\epsilon\_i \sim \mathcal{N}(0,\rho)}[L\_\mathscr{D}(\boldsymbol{w}+\boldsymbol{\epsilon})]$. This condition implies that adding Gaussian perturbations should not reduce the test error, which generally holds for the final solution but not necessarily for all $\boldsymbol{w}$. $w$ represent the parameters influenced by SAM. Applying SAM to the entire network or the entire DFN introduces excessive perturbations, which can hinder DFN’s ability to capture domain-specific information. SAM-SVN strikes a balance by decomposing $w_{DFN}$ and applying perturbations only to the singular value matrix of DFN. The singular value matrix governs the representation space of DFN, thus limiting perturbations to a reasonable spatial range. This prevents DFN from overfitting to the source domain while maintaining its ability to capture domain-specific information during training.
>
> We also quantitatively compared the perturbation on the singular values and other weights in Tab.7.
>
>
>
> ## 3. Comparisons with IFA use a batch size of 96:
>
> IFA is set to a batch size (bsz) of 96, so we adopt the same setting of bsz=96 for comparison to ensure fairness, as stated in appendix section D. Moreover, we also found using different bsz leads to different performance in IFA. For example, for Deepglobe, the performance is 50.1 when bsz=96, while it is 44.7 when bsz=1.
>
>
>
> ## 4. Discussion on GPRN (AAAI'25)：
>
> Since the work had not been published at the time of our submission, we did not include a comparison. We are now offering a discussion on GPRN. GPRN exploits SAM’s generalizability by converting SAM-extracted masks into semantic prompts, aggregating prompt information through graph-based reasoning, and adaptively choosing feedback points. Our approach highlights that adapters naturally function as decouplers. We explore this concept further, proposing a decoupling strategy that is applicable to a variety of models.
>
>
> ## 5. Impact statement in page 9:
>
> Thank you very much for your kind reminder, but the official guidelines indicate that the impact statement is not subject to page limitations.  Here is the original wording: “Papers must be prepared and submitted as a single file: 8 pages for the main paper, **with unlimited pages for references, the impact statement, and appendices.**"

---

### Decision · Program_Chairs · 2025-05-01

**Decision:**

Accept (spotlight poster)

**Comment:**

Hi,

Draft has received overall positive reviews, with 2 weak accept and 2 accept.  Dear authors, please update the draft per recommendations and comments of the reviewers.

Congratulations.

regards

AC